# Genetic structure and *Rickettsia* infection rates in *Ixodes ovatus* and *Haemaphysalis flava* ticks across different altitudes

**Maria Angenica F. Regilme[1,2¤], Megumi Sato[3], Tsutomu Tamura[4], Reiko Arai[4], Marcello Otake Sato[5], Sumire Ikeda[6], Kozo Watanabe[1]** *

1 Center for Marine Environmental Studies (CMES), Ehime University, Matsuyama, Ehime, Japan,
2 Graduate School of Science and Engineering, Ehime University, Matsuyama, Ehime, Japan, 3 Graduate School of Health Sciences, Niigata University, Niigata, Japan, 4 Niigata Prefectural Institute of Public Health and Environmental Sciences, Niigata, Japan, 5 Faculty of Medical Technology, Division of Global Environment Parasitology, Niigata University of Pharmacy and Medical and Life Sciences, Niigata, Japan, 6 Research Laboratories, Research and Development Headquarters, Earth Corporation, Hyogo, Japan

¤ Current address: National Research Center for Protozoan Diseases, Obihiro University of Agriculture and Veterinary Medicine, Obihiro, Hokkaido, Japan
* watanabe.kozo.mj@ehime-u.ac.jp

**Data Availability Statement:** Sequences used for analysis are available in the GenBank database under the accession numbers MW063669 to MW064124, MW065821 to MW066347,

## Abstract

Ixodid ticks, such as *Ixodes ovatus* and *Haemaphysalis flava*, are important vectors of tick-borne diseases in Japan, such as Japanese spotted fever caused by *Rickettsia japonica*. This study describes the *Rickettsia* infection rates influenced by the population genetic structure of *I.ovatus* and *H. flava* along an altitudinal gradient. A total of 346 adult *I. ovatus* and 243 *H. flava* were analyzed for the presence of *Rickettsia* by nested PCR targeting the 17kDA, *gltA*, *rOmpA*, and *rOmpB* genes. The population genetic structure was analyzed utilizing the mitochondrial cytochrome oxidase 1 (*cox1*) marker. The *Rickettsia* infection rates were 13.26% in *I. ovatus* and 6.17% in *H. flava*. For *I. ovatus*, the global $F_{ST}$ value revealed significant genetic differentiation among the different populations, whereas *H. flava* showed non-significant genetic differentiation. The *cox1 I. ovatus* cluster dendrogram showed two cluster groups, while the haplotype network and phylogenetic tree showed three genetic groups. A significant difference was observed in *Rickettsia* infection rates and mean altitude per group between the two cluster groups and the three genetic groups identified within *I. ovatus*. No significant differences were found in the mean altitude or *Rickettsia* infection rates of *H. flava*. Our results suggest a potential correlation between the low gene flow in *I. ovatus* populations and the spatially heterogeneous *Rickettsia* infection rates observed along the altitudinal gradient. This information can be used in understanding the relationship between the tick vector, its pathogen, and environmental factors, such as altitude, and for the control of tick-borne diseases in Japan.

OR975837 to OR975875 and OR975876 to OR975898.

**Funding:** This study was supported by the Ministry of Education, Culture, Sports, Science and Technology, Japan (MEXT) to a project on Joint Usage/Research Center– Leading Academia in Marine and Environment Pollution Research (LaMer).The funders had no role in study design, data collection and analysis, decision to publish, or preparation of the manuscript.

**Competing interests:** The authors have declared that no competing interests exist.

## Introduction

Tick-borne diseases are a significant public health concern in Japan and are transmitted by a diverse range of tick species, such as *Ixodes ovatus* [1] that potentially transmit *Borrelia sp*. causing *Lyme* disease [2] and *Haemaphysalis flava* [3], which transmits *Rickettsia japonica* and is a suspected vector of severe fever with thrombocytopenia syndrome virus [4–6]. Their dispersal is linked to the mobility of their hosts, relying on them to disperse into new landscapes and potentially introduce pathogens [7, 8]. The dynamics of tick-borne pathogens are influenced by the habitat distribution and dispersal behaviors of vectors and hosts along environmental gradients [9]. Therefore, understanding the complex interaction between these factors is important in understanding the spread of tick-borne diseases in Japan.

Tick population genetic analysis provides data that help identify the dispersal pattern of ticks based on gene flow between local populations [10]. The potential of spreading pathogens might be influenced by ticks' dispersal, which is related to the movements of their vertebrate hosts, especially in three-host Ixodidae species [7, 8]. For example, contrasting patterns in the population genetic structures of *I. ovatus* and *H. flava* in the Niigata Prefecture of Japan suggest that host mobility during the immature stages of tick development may influence the genetic structure of adult ticks by affecting survivability into their adult stages [11, 12]. *Ixodes ovatus* populations had greater genetic divergence possibly due to the limited dispersal of their small mammalian hosts during the immature development stage; *H. flava* populations showed a more homogenized structure possibly due to the larger mobility of their large mammalian hosts and avian-mediated dispersal [11]. Other studies have also revealed low gene flow in ticks with low-mobility hosts (e.g., small mammals) and higher gene flow in ticks with highly mobile hosts (e.g., large mammals and birds) [10, 13–16].

The spatial distribution and movement of the vector (i.e., ticks) may determine the spatial distribution of the pathogen's (i.e., *Rickettsia*) infection rate [17]. Previous studies have shown that the pathogen infection rate can be influenced by many factors, such as the vector's genetic diversity, gene flow, and spatial structure [5, 6, 11, 18–25]. For example, previous studies have shown that strong gene flow between local vector populations tends to reduce the spatial heterogeneity of pathogen infection rates between populations [26, 27]. Thus, the spatial distribution and movement of the vector may affect the spatial distribution of the pathogen. To our knowledge, no previous studies have examined the relationship between the spatial heterogeneity of *Rickettsia* infection rates and population genetic structure in ticks.

Environmental factors may relate to the population genetic structure of ticks [28, 29], with limited gene flow increasing genetic variation between populations along an altitudinal gradient, as reported in several studies on other species [30–32]. In the study by [11], no significant influence of environmental factors, including altitude, was observed in the genetic structures of *I. ovatus* and *H. flava* based on the mantel test, but the study did not use any other robust analytical methods to thoroughly examine the influence of altitude on tick genetic structure. In another study, major spotted fever group *Rickettsia* (SFGR) prevalence was analyzed in a total of 3,336 immature and adult ticks across the Niigata Prefecture, Japan in the following tick species: *Dermacentor taiwanensis*, *H. flava*, *Haemaphysalis hystricis*, *Haemaphysalis longicornis*, *Haemaphysalis megaspinosa*, *Ixodes columnae*, *Ixodes monospinosus*, *Ixodes nipponensis*, *Ixodes ovatus*, and *Ixodes persulcatus* [6]. Three SFGR species namely *Rickettsia asiatica*, *R. helvetica* and *R. monacensis* were detected in *H. flava*, *Haemaphysalis longicornis*, *Ixodes monospinus*, *Ixodes nipponensis*, and *Ixodes ovatus*, no spatial distribution of *Rickettsia* infection rates was found among the local populations. To our knowledge, no previous studies have considered the influence of environmental factors on the spatial distribution of Spotted fever group

*Rickettsia* infection rates along an altitudinal gradient in local Ixodid tick populations such as *Ixodes ovatus* and *Haemaphysalis flava* as influenced by the tick population's genetic structure.

In this study, we elucidate the relationship between *Rickettsia* infection rates as influenced by population genetic structure along an altitudinal gradient to improve public health understanding of the distribution of ticks and tick-borne diseases. Based on the isolation by environment (IBE) theory, genetic differentiation increases with environmental variation, regardless of geographic distance [33–35]. Thus based on the results of [11], we hypothesized that in *I. ovatus* with a strong population genetic structure, we expect to see a heterogenous *Rickettsia* infection rate along an altitudinal gradient. In contrast to the homogenous genetic structure of *H. flava* wherein we expect to observe a homogenous *Rickettsia* infection rate.

## Materials and method

### Published data of [6, 11]

In this study, we used *cox1* sequence data from [11] for *I. ovatus* (n = 307) and *H. flava* (n = 220) ticks collected from April 2016 to November 2017 from 30 sites across the Niigata Prefecture, Japan. Sequences used for analysis are available in the GenBank database under the accession numbers MW063669-MW064124 and MW065821—MW066347. *Rickettsia* infection rate data were obtained from [6] from *I. ovatus* (n = 29) and *H. flava* (n = 2), from 38 sites across Niigata Prefecture. The 38 sites surveyed in the previous study by [6] include the 30 sites that were used in this present study (S1 Table). Please refer to [6, 11] for more information about the study sites, collection, sampling identification, DNA extraction, PCR amplification, and sequencing methods used in each respective study.

To strengthen our analysis, we also added new *cox1* sequences and *Rickettsia*-infected/uninfected ticks from *I. ovatus* (n = 39) and *H. flava* (n = 23) individuals collected from April to October 2018, a total of (n = 62) sampled at 30 sites across the Niigata Prefecture, including two sites not previously sampled by [11]. The sequences are available in the GenBank database under the accession numbers OR975837 to OR975875 and OR975876 to OR975898. At these sites, ticks were collected 2–14 times from six core sites among the 30 sites, while ticks were collected once at the remaining sites. The altitude at each site ranged from 8 to 1402 meters above sea level (m.a.s.l.), with a mean altitude of 348 m.a.s.l.

### Unpublished data from the 2018 collection

Ticks collected were stored at 4˚C in microcentrifuge tubes with 70% ethanol. Each collected tick was morphologically identified using a stereo microscope following the identification keys of [36]. Genomic DNA was extracted from individual ticks using Isogenome DNA extraction kits (Nippon Gene Co. Ltd. Tokyo, Japan) following the manufacturer's recommended protocol.

In this study, we combined previously published data from [6] with our newly collected data to calculate the *Rickettsia* infection rate, which is the percentage of *Rickettsia*-infected ticks from each obtained population. We analyzed the obtained tick DNA for spotted fever group *Rickettsia* (SFGR) detection and host identification and amplified the mitochondrial gene *cox1* for population genetic analysis. We performed nested PCR targeting the following genes for the detection of *Rickettsia* sp.: 17-kDA antigen gene (17-kDA); citrate synthase gene (gltA); spotted fever group (SFG)-specific outer membrane protein A gene (rOmpA); and outer membrane protein B gene (rOmpB) as described and analyzed in [6, 37–41] (S2 Table). Briefly, we first amplified the 17-kDa protein. If the results were positive, then PCR was performed to target gltA. Samples that were positive with both 17-kDA and gltA were regarded as positive for SFGR and a nested PCR was performed to target the rOmpA and rOmpB gene,

samples that are positive for 17-Kda, gltA, rOmpA, and rOmpB genes were sequenced to identify the *Rickettsia* species. The amplified PCR products were purified using AMPure XP (Beckman Coulter Co., Japan) and sequenced using the Big Dye Terminator Cycle Sequence Kit (Thermo Fisher Scientific).

The *cox1* mitochondrial gene was amplified by PCR for *cox1* (658 base pairs) using the primer pairs LCO-1490 (5′-GGTCAACAAATCATAAAGATATTGG-3′) and HCO1–2198 (5′-AAACTTCAGGGTGACCAAAAAATCA-3) for phylogenetic analysis and tick species identification [42]. The PCR amplification profile included an initial denaturation of 94°C for 2 min, followed by denaturation at 94°C for 30 s, then annealing at 38°C for 30 s, followed by an extension of 72°C for 1 min for 30 cycles, and a final extension of 72°C for 10 min. The obtained PCR products were purified using the QIAquick 96 PCR Purification Kit (Qiagen, Germany) following the manufacturer's instructions and were sequenced by Eurofin Genomics, Inc. (Tokyo, Japan).

Each forward and reverse read was assembled using CodonCode Aligner version 1.2.4 software (https://www.codoncode.com/aligner/). Low-quality bases were removed in the aligned sequences, and no ambiguous bases were detected. We used the MAFFT alignment online program (https://mafft.cbrc.jp/alignment/server/) to perform multiple alignments using the default settings. The sequences were checked for similarities with the deposited reference sequences from GenBank for sequence quality and tick species confirmation using BLAST (https://blast.ncbi.nlm.nih.gov/Blast.cgi). The protein-coding genes were translated to amino acids to confirm the absence of stop codons and the final aligned sequences were checked in Mesquite version 3.5 [43].

## Population genetic analysis

Multiple sites that are within 80 kilometers were combined for population genetic analysis if less than eight individuals were obtained per site, which resulted in 8 populations labeled A to H (S1 Table). Three sites were excluded from the population genetic analysis because of the limited number of obtained individuals (<8) and the lack of a nearby site within 80 kilometers to combine into a single population.

The final *cox1* sequences of the tick species: *I. ovatus* and *H. flava* were individually analyzed using DNASp version 6.12.03 to determine the haplotype diversity per species [44]. The level of genetic divergence between each population was quantified per species using global $F_{ST}$. Significance was tested using Arlequin software version 3.5.2.2 [45] with 9999 permutations.

The genetic relationship between the *I. ovatus* populations was visualized using the unweighted pair group with the arithmetic mean (UPGMA) cluster method using the APE package [46] for the RStudio software (R Development Core Team, 2016). A cluster dendrogram was created using pairwise $F_{ST}$ values genetic distance matrix from GenAlEx.

## Haplotype network and phylogenetic analyses

We constructed a haplotype network analysis using PopART program version 1.7 (http://popart.otago.ac.nz/index.shtml) on *cox1 I. ovatus* and *H. flava* sequences to assess haplotype relationships and the distribution of *Rickettsia* infected infected ticks using the median-joining network algorithm [47]. Briefly, we constructed a Bayesian phylogenetic tree of *cox1* haplotypes for *I. ovatus* and *H. flava*, respectively, using Markov chain Monte Carlo (MCMC) approach implemented in the BEAST version 1.10.14 [48]. We used the Hasegawa-Kishino-Yano substitution model with estimated base frequencies. We employed a strict clock model and used the coalescent prior as the tree prior. A maximum clade credibility tree was acquired

using TreeAnnotator version 1.10.14 using trees from BEAUti version1.10.14 with 90% of the trees as the burn-in. We viewed the constructed maximum clade credibility tree using FigTree version 1.4.4.

### Statistical analysis

To determine whether there was a significant difference in the *Rickettsia* infection rate between haplotype groups for *I. ovatus* and *H. flava*, we performed a z-score test at $p < 0.05$. The z-score test was chosen because of the large sample size and because the population variance was known. To determine whether there were differences in the mean altitude between the haplotype groups, we used the Welch t-test at $p < 0.05$. Welch t-test was used when the means of the two populations were normally distributed and had equal variances.

### Results

The total number of positive (pos) and negative (neg) ticks for *Rickettsia* infections from the ticks collected in 2018 were: *I. ovatus* (neg = 22, pos = 17) and *H. flava* (neg = 10, pos = 13). In this study, the total number of samples from the previously published data [6, 11] and the unpublished data from 2018 were: *I. ovatus* (n = 346) and *H. flava* (n = 243). The number of adult ticks whose *cox1* was successfully sequenced per species were: *I. ovatus* (346) and *H. flava* (243) (Table 1).

We detected SFGR in 78 (12.44%) out of 627 ixodid ticks, with the highest detected in *I. ovatus* (46/346; 13.29%) and in *H. flava* (15/243; 6.17%) as summarized in Table 1. Out of the 46 *Rickettsia*-infected *I. ovatus* ticks, 25 displayed a 100% identity match with *Rickettsia asiatica* in the 17kDA, *gltA*, and *rOmpB* markers [6], while an additional 19 adult *I. ovatus* from the 2018 collection were also positive with *R. asiatica*. Two haplotypes were found in the rOmpB and 17kDA markers, respectively. One haplotype was found in only one individual (17369). Two out of the 15 *Rickettsia*-infected *H. flava* ticks were found to have the same haplotypes in the 17Kda, gltA,and rOmpA markers, and were identified as *Rickettsia sp.* (LC461063). The remaining 13 *Rickettsia*-infected *H. flava* ticks were identified as *Rickettsia sp*.

Based on the population genetic analysis of the *cox1* sequences, we found a significant global $F_{ST}$ of 0.4154 at $p < 0.05$ for *I. ovatus* (Table 1). In contrast, no significant global $F_{ST}$ values were observed in the *H. flava* (Table 1). There were 59 and 66 *cox1* haplotypes found among the 346 *I. ovatus* and 243 *H. flava* individuals, respectively.

The *cox1* haplotype network of *I. ovatus* (Fig 1) revealed four genetic groups, wherein three genetic groups (1, 2, and 3) were distributed along different altitudinal gradients, as shown in Fig 2. These four genetic groups were concordant with the four clusters found in the *I. ovatus* phylogenetic tree (S1 Fig). The habitat distribution of genetic group 3 was limited to high altitude sites only (range = 16–912 m.a.s.l.), whereas genetic groups 1 (255–471 m.a.s.l.) and 2

**Table 1. Summary of the haplotype and *Rickettsia* infection rates among the 7 ixodid tick species obtained in the Niigata Prefecture, Japan.**

| Tick species ** | ns | n | nh | r | Global $F_{ST}$ |
|---|---|---|---|---|---|
| 1. *Ixodes ovatus* | 30 | 346 | 59 | 46 (13.26%) | 0.4154* |
| 2. *Haemaphysalis flava* | 18 | 243 | 66 | 15 (6.17%) | 0.3597 |
| **Total** | | 589 | | 61 | |

Abbreviations: ns no. of sampling sites; n sample size; nh no of haplotypes; r *Rickettsia* infection rate per species

*p < 0.05 **tick species identification is based on molecular identification using the *cox1* marker and BLAST results

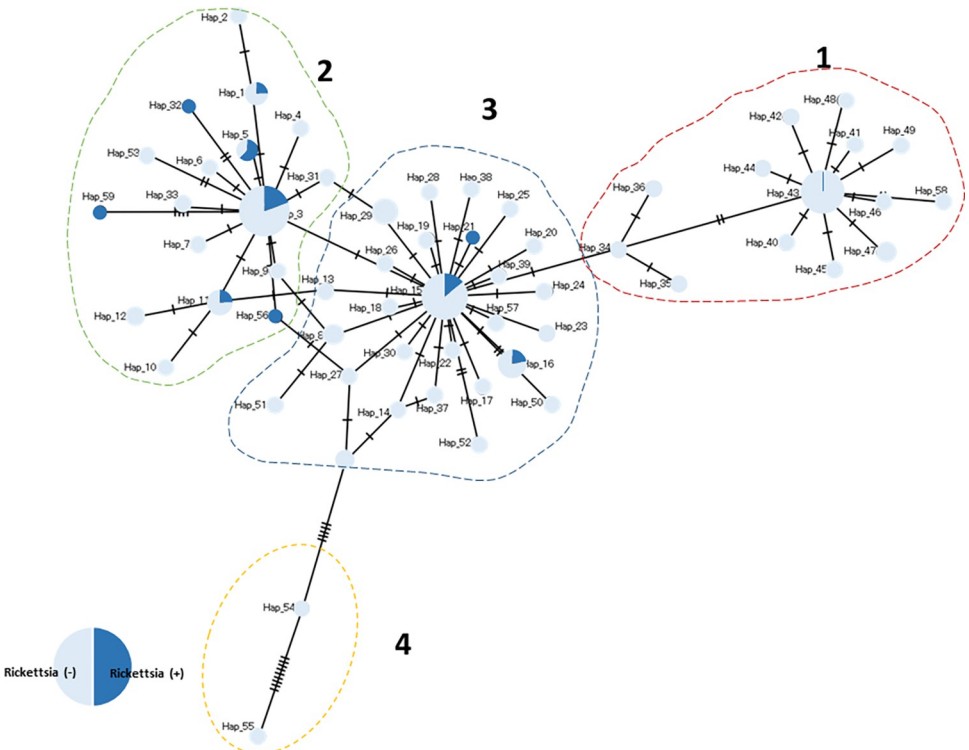

**Fig 1. Median-joining network of the 59 *cox1* haplotype sequences of *Rickettsia* positive and negative *I. ovatus*.** Haplotype groups are indicated by numbers (1 to 4).

(84–1354 m.a.s.l.) were distributed at lower altitudes (Fig 2). The Welch t-test revealed a significant difference between the mean altitudes of genetic groups 1 and 3 at $p < 0.05$ (Table 2); however, no significant difference was observed between genetic groups 1 and 2 or groups 2 and 3. We found a significant difference in the *Rickettsia* infection rates between *I. ovatus* genetic groups 1 and 2 based on the z-score test, but no significant difference between groups 1 and 3 or groups 2 and 3 (Table 2). The mean altitude between the *Rickettsia*-infected (= 273.72 m.a.s.l.) and non-infected *I. ovatus* (= 369.61 m.a.s.l.) revealed a significant difference based on the Welch t-test at $p < 0.05$ (Fig 3). The UPGMA dendrogram of *I. ovatus* revealed two genetic clusters, 1 and 2 using the genetic distance among the seven populations excluding one population due to the limited number of samples (Fig 4).

The cox1 haplotype network of *H. flava* displayed two genetic groups (S2 Fig) consistent with the *H. flava* phylogenetic tree (S3 Fig). No significant difference between the *Rickettsia* infection rates of *H. flava* genetic groups 1 and 2 was observed using the z-score test at $p < 0.05$ (S3 Table).

## Discussion

Our findings support our hypothesis that a genetically structured tick population, such as *I. ovatus* is associated with the *Rickettsia* infection rate to be spatially heterogenous due to limited gene flow along an altitudinal gradient. Our results were consistent with our previous study [11] which suggested that the low mobility of the host species for immature *I. ovatus* contributed to low gene flow in the tick populations. Despite the addition of new samples of *I. ovatus* and *H. flava*, we found a similar pattern of population genetic structure from the

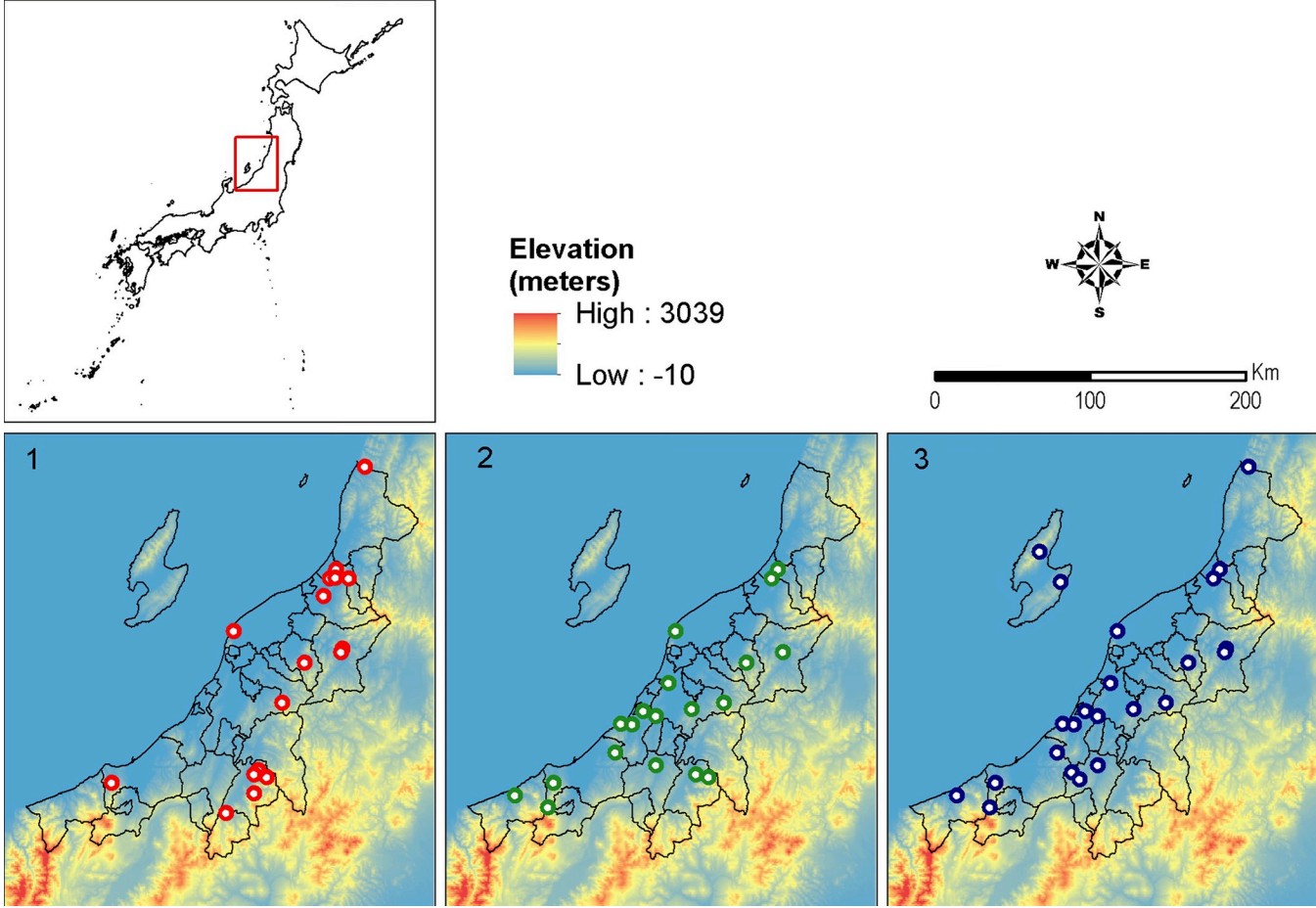

**Fig 2. The influence of altitude on the habitat distribution of *I. ovatus*.** The points indicate the sampling sites of the collected individuals per *I. ovatus* haplotype group (n = 3) in the haplotype network (Fig 1). The map shows the elevation level across the sampling area, with Niigata Prefecture Japan depicted as a color gradation. Group 4, not shown in this figure, with haplotypes 54 and 55 was found on Sado island, which is encircled in black in box 3.

previous study of [11] thus supporting the robustness of their population genetic structure results. The low *I. ovatus* gene flow along the altitudinal gradient might have caused the spatial heterogeneity of *Rickettsia* infection rates among these populations, which is supported by the

**Table 2. The differences in *Rickettsia* infection rates and mean altitude in *I. ovatus* haplotype groups and cluster dendrogram groups.** The table shows the distribution of *Rickettsia*-infected and uninfected *I. ovatus* and the mean altitude in each of the haplotype groups as shown in Fig 1 (*I. ovatus* haplotype network) and the cluster dendrogram in Fig 4 (*I. ovatus* cluster dendrogram). The z-score test showed a significant difference at $p < 0.05$ between the *Rickettsia* detection rates in haplotype groups 1 and 2 (indicated by [ab]) and in cluster dendrogram groups 1 and 2. The Welch t-test at $p < 0.05$ revealed a significant difference in the mean altitude of haplotype groups 1 and 3 and cluster dendrogram groups 1 and 2 indicated by [ab]. Haplotype group 4 was not included in the analysis due to its low sample size.

| Haplotype group | *Rickettsia* infection rate | | | Mean altitude |
|---|---|---|---|---|
| | Positive | Negative | Detection rate | |
| 1 | 1 | 92 | 1.07%[ab] | 294[ab] |
| 2 | 28 | 99 | 22.04%[ab] | 341[a] |
| 3 | 16 | 110 | 12.70% [a] | 374[ab] |
| **Dendrogram Group** | | | | |
| 1 | 4 | 120 | 3.23% [ab] | 30 |
| 2 | 40 | 227 | 14.98% [ab] | 345 |

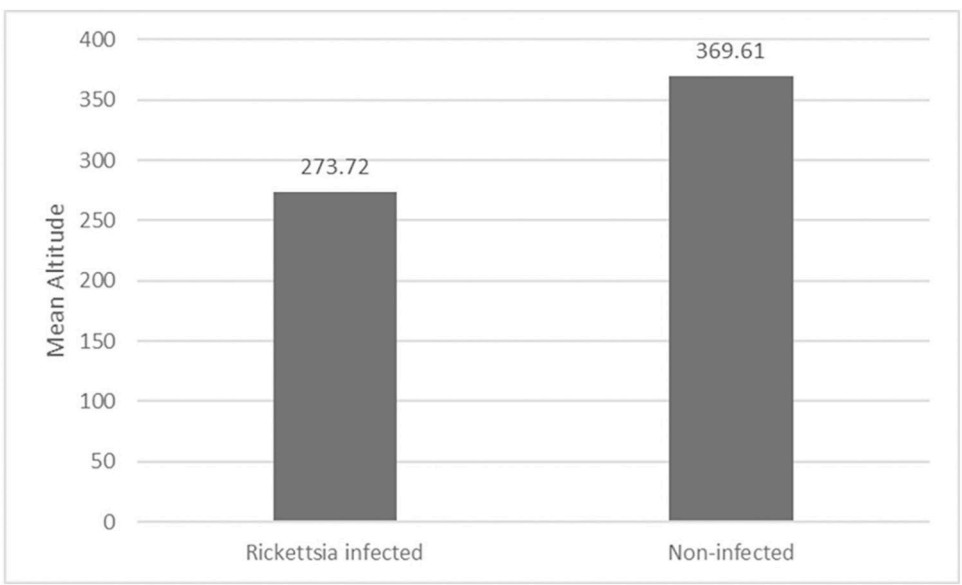

**Fig 3. The relationship between the mean altitude of *Rickettsia* positive (n = 46) and negative (n = 300) *I. ovatus*.** Welch t-tests revealed a significant difference in the mean altitude between *Rickettsia-infected* and non-infected *I. ovatus* at $p < 0.05$. Populations with combined sites are labeled A to H.

significant difference found in *Rickettsia* infection rates between genetic clusters 1 and 2. A similar pattern was observed in the studies of [49–51] which found that the infection rate of *Borrelia burgdorferi*, the causative agent of Lyme disease, decreased in ticks along the

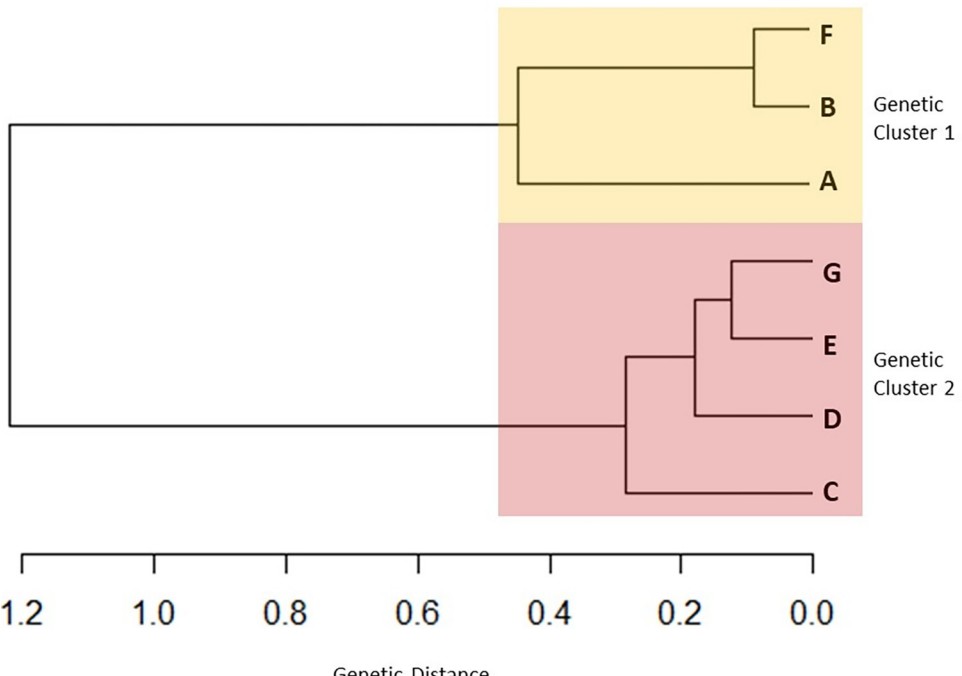

**Fig 4. An unweighted pair group method with the arithmetic mean (UPGMA) dendrogram of *I. ovatus* based on the pairwise genetic distance ($F_{ST}$) of *cox1* among the 7 populations across Niigata Prefecture, Japan.** We excluded one population due to the limited number of samples.

altitudinal gradient. Low gene flow can cause infected and uninfected ticks to have limited opportunities to traverse a wider spatial area thus causing a heterogeneous *Rickettsia* infection rate [52, 53].

We found two genetic groups in the *H. flava* haplotype network, but no significant difference in the *Rickettsia* infection rates between the two groups. These results might be due to the high gene flow observed in the *H. flava* populations, which enable *Rickettsia*-infected and uninfected *H. flava* individuals to traverse between the study sites. The high mobility of the large mammalian hosts used by adult *H. flava* and avian-mediated dispersal during their immature stage probably contributed to their homogenized population genetic structure [11], and the resulting homogenized *Rickettsia* infection rates. Large mammalian hosts and birds have a wide dispersal range that enables the broader movement of *Rickettsia*-infected ticks, as observed in previous studies of *Amblyomma Americanum* [13, 14, 54], *H. flava* [11], and *I. ricinus* [55]. Birds are especially good at dispersing over large areas since they can easily traverse landscape barriers such as mountains, fences, glaciers, and oceans that would be difficult for mammals to cross [56].

The different *Rickettsia* infection rates and altitudinal ranges between the *I. ovatus* phylogenetic groups maybe caused by diverse factors such as host availability and distribution, other environmental factors such as climate and vegetation, and anthropogenic factors such as urbanization. However, the adaptive evolutionary theory, which states that organisms adjust to new or severe changes in their environment to become better suited to their habitat [57, 58] maybe the best explanation for our results. Based on the relationship between the *I. ovatus* phylogenetic groups and their mean attitudes, *I. ovatus* might be undergoing local adaptation along the altitudinal gradient due to the higher genetic differentiation between populations as supported by the significant global $F_{ST}$ (0.4154) found in *I. ovatus*. Based on isolation by environment (IBE), genetic differentiation will increase with increased environmental differences independent of geographic distances [33, 34, 59]. Thus in our study, the addition of *I. ovatus* together with the published data of [11] collected from an altitudinal gradient have shown genetic differences and different *Rickettsia* infection rate. When environmental conditions differ, the success of immigration in a new habitat is reduced, which may increase the genetic fixation rate due to a lower chance of outcrossing; thereby enhancing genetic isolation [60]. Thus, the lower gene flow along the altitudinal gradient reduced the spatial homogeneity of *Rickettsia* infection rates among the *I. ovatus* tick populations, thus causing the different *Rickettsia* infection rates obtained.

The occurrence of local adaptation in tick populations could affect the future of the tick-borne disease landscape. Environmental factors, such as precipitation, temperature, and altitude, have been shown to drive population differentiation in insects, such as *Anopheles* mosquitoes and *Drosophila* flies [61–63], but studies on the environmental adaptation of ixodid ticks, such as *I. ovatus*, and its *Rickettsia* infection susceptibility, have not yet been performed. [7] suggested that environmental conditions that affect bird hosts can also affect the local adaptation of ticks. Few studies have assessed such local adaptation in multiple organisms with varying dispersal abilities [64–68] and is an area in need of future research.

One of the limitations of this study is the use of one mitochondrial gene *cox1* which limited us to compare our results with other target genes to highly support our findings. If markers with high mutation rates or many markers were used, it might have been possible to look at even finer population genetic structure and see differences in infection rates among the subdivided populations. Despite this, we were able to determine the relationship between the tick population genetic structure and *Rickettsia* infection rates as influenced by the altitudinal gradient. The mitochondrial *cox1* gene has been widely used for population genetic analysis of many tick species and was proven to be informative in determining the relationship from the

subfamily to the population levels [69–73]. Mitochondrial genes have a mutation rate that is useful in species-level phylogenetics and can be used for wide geographic ranges however its resolution is not fine enough to study species selection [10]. In future studies, we suggest including additional mitochondrial genes and or nuclear genes.

Since ticks are blood-sucking ectoparasites, they directly influence their mammalian hosts and the pathogens they transmit [74–76]. The interaction between the vector (tick), host, and pathogen (*Rickettsia*) is essential in understanding and predicting the risk and transmission of tick-borne diseases [77]. Understanding the genetic structure of ticks can serve as an alternative indicator to infer the potential spread of its pathogen [78]. Our study found relationships between (1) the population genetic structure of ticks and the corresponding *Rickettsia* infection rates, (2) altitude and the population genetic structure of ticks, and (3) altitude and *Rickettsia* infection rates. Though our results can provide a useful information about the tick distribution and possible potential spread of pathogens, there are some factors that should also be considered to apply our results such as ticks can have different mammalian hosts during different life stages in the field that have varying hosts mobility and other environmental factors can also affect such as temperature, humidity etc.. can also be a factor. We found that host mobility may influence the genetic structure of ixodid ticks. This information can be used to design more effective tick-borne disease control programs that focus on screening and detecting pathogens found in ticks and their mammalian hosts. For example, patterns of disease transmission from ticks with a high genetic divergence and less mobile hosts, such as *I. ovatus*, are likely due to the movement of infected hosts rather than infected ticks. Thus, screening prospective tick hosts for *Rickettsia* infection would be more suitable in this example. We suggest screening the hosts of immature *I. ovatus*, such as small rodents, instead of screening ticks. The *Rickettsia* infection rate in tick genetic groups can predict the spread of tick-borne diseases caused by *Rickettsia*, such as the Japanese spotted fever. We also found that altitude may influence the *Rickettsia* infection rate of *I. ovatus* genetic groups. This information can be used to determine the high-risk areas (e.g., lowland, mountains, etc.) of tick-borne diseases along an altitudinal gradient. Genetically structured arthropod vectors, such as ticks, can have different vector competencies, and environmental factors, such as altitudinal gradients, that can influence the vector's ability to acquire, transmit, and maintain the pathogen infection [79].

## Supporting information

**S1 Table. Summary of *Ixodes ovatus* and *Haemaphysalis flava* collected from the different locations of Niigata Prefecture and its corresponding sample number and number of *Rickettsia* infection per site.**
(XLSX)

**S2 Table. Summary of PCR primers used in the detection of Spotted fever group *Rickettsia*.**
(DOCX)

**S3 Table. The difference in *Rickettsia* infection rates in *H. flava* haplotype groups and a table showing the distribution of *Rickettsia*-infected and uninfected *H. flava* in each genetic groups as shown in S2 Fig (*H. flava* median-joining network).** The results of the z-score test for two populations proportions at $p < 0.05$ showed no significant difference between the *Rickettsia* detection rate in haplotype groups 1 and 2. The Welch t-test at $p < 0.05$ revealed no significant difference in the mean altitude of the two groups.
(XLSX)

**S1 Fig. Phylogenetic tree from the BEAST analysis of 59 haplotype *cox1* sequences of *I. ovatus*.** The blue-labeled haplotypes indicate the presence of *Rickettsia* infection. The red

parentheses provide the number of *Rickettsia*-positive individuals per haplotype. The black labeled haplotypes are negative for *Rickettsia* infection.
(DOCX)

**S2 Fig. Median-joining network of the 66** *cox1* **haplotype sequences of** *Rickettsia* **positive and negative** *H. flava.* Haplotype groups are indicated as 1 and 2.
(DOCX)

**S3 Fig. Phylogenetic tree from the BEAST analysis of 66 haplotype** *cox1* **sequences of** *H. flava.* The blue-labeled haplotypes indicate *Rickettsia* infection in individual samples. The parentheses in red provide the number of *Rickettsia*-infected ticks. The black-labeled haplotypes are negative for *Rickettsia* infection.
(DOCX)

## Acknowledgments

The authors are thankful for the assistance from the Niigata Prefectural Office during the tick sampling collection. We would also like to thank the alumni of MECOH Lab Ehime University for their help in molecular analyses: Masaya Doi, Kohki Tanaka, and Mizuki Ueda. We are also grateful to Micanaldo Francisco for the construction of the map for this manuscript. We would like to thank Enago (www.enago.jp) for the English language review and proofreading of the manuscript.

## Author Contributions

**Conceptualization:** Maria Angenica F. Regilme, Megumi Sato, Tsutomu Tamura, Reiko Arai, Marcello Otake Sato, Sumire Ikeda, Kozo Watanabe.

**Data curation:** Maria Angenica F. Regilme, Megumi Sato, Tsutomu Tamura, Marcello Otake Sato, Sumire Ikeda, Kozo Watanabe.

**Formal analysis:** Maria Angenica F. Regilme, Kozo Watanabe.

**Funding acquisition:** Megumi Sato, Kozo Watanabe.

**Investigation:** Maria Angenica F. Regilme, Tsutomu Tamura, Kozo Watanabe.

**Methodology:** Maria Angenica F. Regilme, Megumi Sato, Reiko Arai, Marcello Otake Sato, Sumire Ikeda, Kozo Watanabe.

**Project administration:** Kozo Watanabe.

**Supervision:** Kozo Watanabe.

**Validation:** Kozo Watanabe.

**Visualization:** Maria Angenica F. Regilme.

**Writing – original draft:** Maria Angenica F. Regilme.

**Writing – review & editing:** Maria Angenica F. Regilme, Megumi Sato, Tsutomu Tamura, Reiko Arai, Marcello Otake Sato, Sumire Ikeda, Kozo Watanabe.

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
