## [Decision Letter · Decision Letter 0]

7 Jun 2023

PONE-D-23-12494Rickettsia infection rate along an altitudinal gradient as influenced by population genetic structure of Ixodid ticksPLOS ONE

Dear Dr. Watanabe,

Thank you for submitting your manuscript to PLOS ONE. After careful consideration, we feel that it has merit but does not fully meet PLOS ONE’s publication criteria as it currently stands. Therefore, we invite you to submit a revised version of the manuscript that addresses the points raised during the review process.

We look forward to receiving your revised manuscript.

Kind regards,

Maria Stefania Latrofa

Academic Editor

PLOS ONE

4. We note that Figure 2 in your submission contain [map/satellite] images which may be copyrighted. All PLOS content is published under the Creative Commons Attribution License (CC BY 4.0), which means that the manuscript, images, and Supporting Information files will be freely available online, and any third party is permitted to access, download, copy, distribute, and use these materials in any way, even commercially, with proper attribution. For these reasons, we cannot publish previously copyrighted maps or satellite images created using proprietary data, such as Google software (Google Maps, Street View, and Earth). For more information, see our copyright guidelines: http://journals.plos.org/plosone/s/licenses-and-copyright.

Additional Editor Comments:

Dear Authors, although the total number of ticks and haplotypes for I. ovatus and H. flava, were slightly increased, the main concepts have already been published (Infect Genet Evol. 2021 Oct;94:104999. doi: 10.1016/j.meegid.2021.104999).

I decided to rate/accept the work as a major revision if previously published data is removed from this paper (see, for example, lines 221-235 and 271-274).

I suggest focusing the article on Rickettsia data, supported by previous results.

Reviewers' comments:

Reviewer's Responses to Questions

**Comments to the Author**

1. Is the manuscript technically sound, and do the data support the conclusions?

Reviewer #1: No

Reviewer #2: Yes

2. Has the statistical analysis been performed appropriately and rigorously? 

Reviewer #1: No

Reviewer #2: Yes

3. Have the authors made all data underlying the findings in their manuscript fully available?

Reviewer #1: No

Reviewer #2: Yes

4. Is the manuscript presented in an intelligible fashion and written in standard English?

Reviewer #1: No

Reviewer #2: Yes

5. Review Comments to the Author

Reviewer #1: The data of the manuscript have been previously published and re-analysis has been performed on them. I think you should just focus on the new main data. Therefore, it is recommended to rewrite the manuscript and also, submit it to an entomology journal.

Reviewer #2: The manuscript is really interesting and brings good results for scientific community. Authors studied the ecological and genetic factors that may affect the infection rate of Rickettsia spp., an important pathogen of zoonotic concern. The study is well designed, results are clear and shows a correlation between genetic structure of ticks and infection rate.

Title: I would suggest (see coment bellow on results) to adjust the title of the manuscript:

Rickettsia infection rate along an altitudinal gradient as influenced by population genetic structure of Ixodes ovatus and Haemaphysalys flava ticks.

Since it was not possible to evaluate the altitudinal influence on the genetic structure of many Ixodid species, it is better to highlight the species present on the study. Also, on the abstract, authors wrote the goal/results of the study being: ‘This study describes the population genetic structure and gene flow of I.ovatus and H. flava and their Rickettsia infection rates along an altitudinal gradient. A total of 346 adult I. ovatus and 243 H. flava were analyzed for the presence of Rickettsia by nested PCR targeting the 17kDA, gltA, rOmpA, and rOmpB genes’, not mentioning the other tick species sampled and tested.

The manuscript could be benefit if some sections were rewrite to make the text more flowing.

For example:

Line 51: 'Tick dispersal of many three-host Ixodid ticks depends on the movements of their vertebrate hosts, which influences each tick’s potential to spread its pathogen [7-8].'

- The word tick is repeated 3 times in one sentence. Suggestion: The potential of spreading pathogens might be influenced by ticks’ dispersal, which is related to the movements of their vertebrate hosts, especially in three-host Ixodidae species.

Introduction

Lines 98-102: ‘The present study used cytochrome oxidase 1 (cox1) mitochondrial gene

sequences and Rickettsia-infected and uninfected data…..

- This is material and methods, and should not be mentioned in the introduction.

Material and Methods

Line 109: I. monospinus (n = 4), and I. nipponensis (n = 2) from…

- I suggest to remove the tick species, since it was not possible to evaluate the

Line 154: The protein-coding genes were translated to amino acids to confirm the absence of stop codons.

- Using which software - citation?

Line 158: Multiple sites that are in proximity to each other were combined for population genetic analysis

- What ‘in proximity’ means? If another research wants to repeat the experiment, how should populations be combined by ‘proximity’? Distance of 1 kilometer, 100 kilometers?

Results

Line 192: I. monospinus (neg = 11, pos = 6); I. asanumai (neg = 4, pos=5); I. nipponensis (pos = 4); I. persulcatus (neg = 4); and H. japonica (pos = 2).

- I would suggest to remove all results from the other tick species, since it does not fit the objective of the study: ‘This study describes the population genetic structure and gene flow of I. ovatus and H. flava and their Rickettsia infection rates along an altitudinal gradient’.

6. PLOS authors have the option to publish the peer review history of their article (what does this mean?). If published, this will include your full peer review and any attached files.

Reviewer #1: No

Reviewer #2: No

---

## [Author Response · Author response to Decision Letter 0]

14 Aug 2023

2023 August 14

Dr. Maria Stefania Latrofa

Academic Editor

PLOS ONE

Dear Dr. Latrofa,

We are pleased to submit the revised version of the submitted manuscript entitled “Rickettsia infection rate along an altitudinal gradient as influenced by population genetic structure of Ixodes ovatus and Haemaphysalis flava ticks”. 

The manuscript was modified accordingly to the editor’s and reviewers’ comments and suggestions. Moreover, the authors answered and explained carefully each of the feedback received from both the editors and reviewers. For your reference, the manuscript was revised accordingly with its corresponding line numbers and highlighted the changes within the manuscript. 

The authors confirmed that this manuscript has not been published or is under consideration in another journal. All the authors have read and approved the revisions and contents of this manuscript and agreed to the submission policies of PLOS ONE. All authors have contributed to the research and manuscript writing. The authors have no conflicts of interest to disclose. 

Moreover, following PLOS ONE’s style requirements and file naming, we have revised the manuscript carefully. 

The authors acknowledge the editors and the reviewers for their helpful comments and suggestions thus improving the overall manuscript. We are looking forward that these revisions will warrant acceptance for publication in your notable journal, PLOS ONE. 

Sincerely Yours, 

Dr. Kozo Watanabe, Professor

Center for Marine Environmental Studies (CMES) 

Ehime University, Bunkyo-cho 3, Matsuyama, 790-8577, Japan 

Email: watanabe.kozo.mj@ehime-u.ac.jp

Editor’s comments: Changes are highlighted in green in the revised manuscript with the track changes file

Response: Thank you very much for this comment. We now revised the manuscript following the PLOS ONE style's requirements. Highlighted in green in the manuscript are the changes for the affiliation format, font size per manuscript section, and mention of Figures were changed to Fig. in adherence to PLOS ONE's style requirements.

Response: Sequences used for analysis are available in the GenBank database under the accession numbers MW063669-MW064124 and MW065821 - MW066347. Additional sequences of I.ovatus (n=39) and H.flava (n=23) collected in 2018 were deposited in NCBI Genbank with the accession numbers currently being processed, we will inform PLOS One immediately once we receive the numbers.

Response: The authors acknowledged this comment thus we deposited on NCBI Genbank the additional sequences of I.ovatus (n=39 ) and H.flava (n=23) currently under submission at NCBI and accessions numbers are being processed, once we receive it we will inform PLOS One immediately.

4. We note that Figure 2 in your submission contain [map/satellite] images which may be copyrighted. All PLOS content is published under the Creative Commons Attribution License (CC BY 4.0), which means that the manuscript, images, and Supporting Information files will be freely available online, and any third party is permitted to access, download, copy, distribute, and use these materials in any way, even commercially, with proper attribution. For these reasons, we cannot publish previously copyrighted maps or satellite images created using proprietary data, such as Google software (Google Maps, Street View, and Earth). For more information, see our copyright guidelines: http://journals.plos.org/plosone/s/licenses-and-copyright.

Response: The authors appreciate your comment about Figure 2. This figure is not a previously copyrighted map or satellite image from proprietary data. The map was created using ArcGIS software version 10.2 (ESRI, Redlands, CA). In creating the map, we used digital elevation data and administrative boundaries data. The digital elevation data is a single-band raster image generated from the Shuttle Radar Topography Mission (SRTM) satellite. STRM data is provided at a spatial resolution of 1 arc-second (approximately 30m) (Farr, et al. 2007). This satellite-based elevation data was freely obtained using Google Earth Engine (GEE) code editor platform (Gorelick, et al. 2017). GEE code editor is a web-based Integrated Development Environment for writing and running Java scripts to support geospatial analysis (Google 2021). The GIS data of administrative boundaries of Japan was freely obtained in vector polyline format from the DIVA-GIS website (DIVA-GIS n.d.). DIVA-GIS is aimed at those who cannot afford generic commercial geographic information system data and software (Hijmans, Guarino, and Mathur 2012). 

5. Dear Authors, although the total number of ticks and haplotypes for I. ovatus and H. flava, were slightly increased, the main concepts have already been published (Infect Genet Evol. 2021 Oct;94:104999. doi: 10.1016/j.meegid.2021.104999).

I decided to rate/accept the work as a major revision if previously published data is removed from this paper (see, for example, lines 221-235 and 271-274).

I suggest focusing the article on Rickettsia data, supported by previous results.

Response: The authors are grateful for this valuable comment. We would like to clarify that 

we agree with the editor's point that the concept of genetic differentiation along elevation has already been published. Therefore, in the discussion of the revised manuscript, we have removed all elements related to the published concepts, focusing primarily on the differences in Rickettsia infection rates observed among genetic groups of I. ovatus along an altitudinal gradient, which is the main and new concepts of this study. However, we could not remove the results of the population genetic structure based on the newly obtained in this study and previously published cox1 data from the result section, as this information is the basis for reporting the new concept of this study. 

Specifically, the following lines in the original manuscript were removed in the revised manuscript.

Original MS, changes are highlighted in green with strikethrough

L284-286: The significant global FST estimate of 0.4154 among the I. ovatus populations revealed genetic differentiation between the populations as supported by the occurrence of two genetic clusters in the cluster dendrogram.

Original MS L309-329 In addition to host mobility, environmental factors can influence the population genetic structure of ticks [54]. Each tick species has preferred environmental conditions that are conducive to completing the tick life cycle, thus influencing the geographical distribution of ticks and the risk areas for tick-borne diseases [55-56]. Altitude may influence the population genetic structure of ticks through the effect of altitude on the distribution and abundance of ticks and/or their hosts [57-59]. Altitudinal differences between populations can affect genetic divergence [60], through, for example, ecological isolation, which causes natural selection against maladapted immigrants and limits gene flow [61-62]. For example, organisms adapted to low altitudinal sites that cannot tolerate the lower temperatures at higher altitudes would not survive if they dispersed to those higher altitude sites, thus restricting gene flow across the altitudinal gradient [60]. 

The difference in mean altitude between I. ovatus cluster groups 1 and 2 might be due to adaptive divergence along the altitudinal gradient. Individuals from cluster group 1 were distributed in higher altitude areas of the northern mountainous area of Niigata Prefecture, while cluster group 2 individuals were found in the lower altitude regions of northern Niigata Prefecture. Populations along an altitudinal gradient are prone to differentiating selection pressures, which result in local adaptation [63]. Altitudinal gradients may also cause the varying ambient temperature, precipitation, and humidity levels essential to ticks' development and survival [64]. These varying environmental factors may cause ticks to have difficulty dispersing over a wide habitat range. Thus, ticks may need to adapt to extreme habitats, such as extreme altitude or precipitation levels, for survival. For example, Rhipicephalus compositus were found in altitudes of 1000–2500 m, but optimal conditions were between 1200–2500 m [64-66].

And the following lines were left in the Revised MS: L221-239:

Based on the population genetic analysis of the cox1 sequences, we found a significant global FST of 0.4154 at p < 0.05 for I. ovatus (Table 1). In contrast, no significant global FST values were observed in the H. flava, I. monospinus, I. asanumai, I. nipponensis, or I. persulcatus samples (Table 1). There were 59 and 66 cox1 haplotypes found among the 346 I. ovatus and 243 H. flava individuals, respectively. We found the following number of cox1 haplotypes in the remaining species: I. monospinus (n = 9), I. asanumai (n = 4), I. nipponensis (n = 4), I. persulcatus (n = 3), and H. japonica (n = 2).

The cox1 haplotype network of I. ovatus (Fig 1) revealed four genetic groups, wherein three genetic groups (1, 2, and 3) were distributed along different altitudinal gradients, as shown in Fig 2. These four genetic groups were concordant with the four clusters found in the I. ovatus phylogenetic tree (S1 Fig). The habitat distribution of genetic group 3 was limited to high altitude sites only (range = 16–912 m.a.s.l.), whereas genetic groups 1 (255–471 m.a.s.l.) and 2 (84–1354 m.a.s.l.) were distributed at lower altitudes (Fig 2). The Welch t-test revealed a significant difference between the mean altitudes of genetic groups 1 and 3 at p < 0.05 (Table 2); however, no significant difference was observed between genetic groups 1 and 2 or groups 2 and 3. 

The following L271-274 were also removed: The cox1 haplotype network of H. flava displayed two genetic groups (S2 Fig) consistent with the H. flava phylogenetic tree (S3 Fig). Based on the Welch t-test, we found no significant differences in mean altitude (205 and 165 m.a.s.l.) at p < 0.05 between the two H. flava genetic groups (S3 Table).

The authors also modified the objectives of the manuscript to highlight the relationship between Rickettsia infection rate and population genetic structure. 

The following lines were removed in the introduction L73-77: Thus, in this study, we expected to see a highly divergent population genetic structure in I. ovatus along an altitudinal gradient due to the limited movement of their mammalian hosts along that gradient; whereas H. flava should show a less divergent structure along the altitudinal gradient due to the higher mobility of its hosts. To our knowledge, no studies have focused on tick gene flow along an altitudinal gradient. 

We also revised the following lines 92-93: Here, we determine the relationship between Rickettsia infection rates as influenced by population genetic structure along an altitudinal gradient.

We also included the following lines 95-98: Thus based on the results of [11], we hypothesized that in I. ovatus with a strong population genetic structure, we expect to see a heterogenous Rickettsia infection rate along an altitudinal gradient. In contrast to the homogenous genetic structure of H. flava wherein we expect to observe a homogenous Rickettsia infection rate. 

The following lines 288-290 were added: Despite the addition of new samples of I. ovatus and H. flava, we found a similar pattern of population genetic structure from the previous study of [11] thus proving the robustness of their population genetic structure results. 

Reviewer 1’s comments: Changes are highlighted in purple in the revised manuscript with the track changes file 

1. Is the manuscript technically sound, and do the data support the conclusions?

No

Response: The authors appreciate your comment. 

2. Has the statistical analysis been performed appropriately and rigorously? No

Response: The authors have performed rigorous statistical analysis to determine the significant difference of the Rickettsia infection rate between the haplotype groups for both species I.ovatus and H. flava, please see L237-241: 

The Welch t-test revealed a significant difference between the mean altitudes of genetic groups 1 and 3 at p < 0.05 (Table 2); however, no significant difference was observed between genetic groups 1 and 2 or groups 2 and 3. We found a significant difference in the Rickettsia infection rates between I. ovatus genetic groups 1 and 2 based on the z-score test, but no significant difference between groups 1 and 3 or groups 2 and 3 (Table 2). The mean altitude between the Rickettsia-infected (=273.72 m.a.s.l.) and non-infected I. ovatus (=369.61 m.a.s.l.) revealed a significant difference based on the Welch t-test at p < 0.05 (Fig 3).

3. Have the authors made all data underlying the findings in their manuscript fully available?

Response: The authors have uploaded the sequences used in the analysis in NCBI repository with accession numbers MW063669-MW064124 and MW065821 - MW066347. Additional sequences of I.ovatus (n=39) and H.flava (n=23) collected in 2018 were deposited in NCBI Genbank and are currently being processed for accession number. We will inform PLOS One soon once we received the accession numbers.

4. Is the manuscript presented in an intelligible fashion and written in standard English?

No

Response: Thank you for your comment. Our manuscript has been checked for English grammar and language style and revised accordingly by a reputable language proofreading company for scientific research articles, Enago.

5. Review Comments to the Author

The data of the manuscript have been previously published and re-analysis has been performed on them. I think you should just focus on the new main data. Therefore, it is recommended to rewrite the manuscript and also, submit it to an entomology journal.

Response: The authors are thankful for this comment. We revised the manuscript accordingly to highlight the Rickettsia data and use the previously published data and additional sequence data for both I. ovatus and H. flava to support our results that low gene flow in the I. ovatus populations has caused spatially heterogenous Rickettsia infection rates along the altitudinal gradient. Please also see our response to the editor’s comment E5.

6. PLOS authors have the option to publish the peer review history of their article (what does this mean?). If published, this will include your full peer review and any attached files.

Do you want your identity to be public for this peer review? For information about this choice, including consent withdrawal, please see our Privacy Policy.

No 

Response: The authors respect the decision of the reviewer to remain anonymous once the manuscript is accepted for publication.

Reviewer 2’s comments: Changes are highlighted in blue in the revised manuscript with the track changes file 

1. Is the manuscript technically sound, and do the data support the conclusions?

Yes 

Response: The authors are grateful for this feedback.

2. Has the statistical analysis been performed appropriately and rigorously? Yes 

Response: Thank you for this comment.

3. Have the authors made all data underlying the findings in their manuscript fully available?

The PLOS Data policy requires authors to make all data underlying the findings described in their manuscript fully available without restriction, with rare exception (please refer to the Data Availability Statement in the manuscript PDF file). The data should be provided as part of the manuscript or its supporting information, or deposited to a public repository. For example, in addition to summary statistics, the data points behind means, medians and variance measures should be available. If there are restrictions on publicly sharing data—e.g. participant privacy or use of data from a third party—those must be specified. Yes 

Response: The authors are thankful for these comments.

4. Is the manuscript presented in an intelligible fashion and written in standard English?

PLOS ONE does not copyedit accepted manuscripts, so the language in submitted articles must be clear, correct, and unambiguous. Any typographical or grammatical errors should be corrected at revision, so please note any specific errors here. Yes

Response: Thank you for your time and effort to read and give valuable comments for the improvement of the manuscript.

5. Review Comments to the Author

The manuscript is really interesting and brings good results for scientific community. Authors studied the ecological and genetic factors that may affect the infection rate of Rickettsia spp., an important pathogen of zoonotic concern. The study is well designed, results are clear and shows a correlation between genetic structure of ticks and infection rate.

Response: The authors are thankful for Reviewer 2’s comments and suggestions to improve the manuscript. 

6. I would suggest (see coment bellow on results) to adjust the title of the manuscript:

Rickettsia infection rate along an altitudinal gradient as influenced by population genetic structure of Ixodes ovatus and Haemaphysalys flava ticks. Since it was not possible to evaluate the altitudinal influence on the genetic structure of many Ixodid species, it is better to highlight the species present on the study. Also, on the abstract, authors wrote the goal/results of the study being: ‘This study describes the population genetic structure and gene flow of I.ovatus and H. flava and their Rickettsia infection rates along an altitudinal gradient. A total of 346 adult I. ovatus and 243 H. flava were analyzed for the presence of Rickettsia by nested PCR targeting the 17kDA, gltA, rOmpA, and rOmpB genes’, not mentioning the other tick species sampled and tested.

The manuscript could be benefit if some sections were rewrite to make the text more flowing.

Response: We appreciate your suggestion and agreed with it. We changed the title, please see L1-2: 

Rickettsia infection rate along an altitudinal gradient as influenced by population genetic structure of Ixodes ovatus and Haemaphysalis flava tick

Please see L23-24:

This study describes the Rickettsia infection rates influenced by the population genetic structure of I.ovatus and H. flava along an altitudinal gradient. 

7. Line 51: Tick dispersal of many three-host Ixodid ticks depends on the movements of their vertebrate hosts, which influences each tick’s potential to spread its pathogen [7-8].'

- The word tick is repeated 3 times in one sentence. Suggestion: The potential of spreading pathogens might be influenced by ticks’ dispersal, which is related to the movements of their vertebrate hosts, especially in three-host Ixodidae species.

Response: The following L50-52 have been revised accordingly: 

The potential of spreading pathogens might be influenced by ticks’ dispersal, which is related to the movements of their vertebrate hosts, especially in three-host Ixodidae species [7-8].

8. Lines 98-102 ‘The present study used cytochrome oxidase 1 (cox1) mitochondrial gene

sequences and Rickettsia-infected and uninfected data…..

- This is material and methods, and should not be mentioned in the introduction.

Response: Thank you for your valuable comment. We removed the following L97-101 

in the introduction of the original MS and incorporated it in the materials and methods section of the revised MS L105-106:

In this study, we used cox1 sequence data from [11] for I. ovatus (n = 307) and H. flava (n = 220) ticks collected from April 2016 to November 2017 from 30 sites across the Niigata..... and L113-116 To strengthen our analysis, we also added new cox1 sequences and Rickettsia-infected/uninfected ticks from I. ovatus (n= 39) and H. flava (n=23) individual collected from April to October 2018 , total of (n=62) sampled at 30 sites across the Niigata Prefecture, including two sites not previously sampled by [11]. 

9. Line 109 I. monospinus (n = 4), and I. nipponensis (n = 2) from…

- I suggest to remove the tick species, since it was not possible to evaluate the 

Response: Thank you for the suggestion. We would like to clarify what is not possible to evaluate? Since the comment was incomplete, the authors are unsure of the whole meaning of the comment however we removed the following L108: I. monospinus (n = 4), and I. nipponensis (n = 2).

Please see L109-110 of the revised MS: Rickettsia infection rate data were obtained from [6] from I. ovatus (n = 29) and H. flava (n = 2) from 38 sites across Niigata Prefecture. 

10. Line 154 The protein-coding genes were translated to amino acids to confirm the absence of stop codons.

- Using which software - citation?

Response: We revised it accordingly, please see L156-158: 

The protein-coding genes were translated to amino acids to confirm the absence of stop codons and the final aligned sequences were checked in Mesquite version 3.5 [38]. 

11. Line 158: Multiple sites that are in proximity to each other were combined for population genetic analysis

- What ‘in proximity’ means? If another research wants to repeat the experiment, how should populations be combined by ‘proximity’? Distance of 1 kilometer, 100 kilometers?

Response: We revised the sentence accordingly. Please see L161-162:

Multiple sites that are within 80 kilometers were combined for population genetic analysis if less than eight individuals were obtained per site, which resulted in 8 populations labeled A to H (S1 Table).

In this regard, we also added information about this in the following L163-165: 

Three sites were excluded from the population genetic analysis because of the limited number of obtained individuals (<8) and the lack of a nearby site within 80 kilometers to combine into a single population.

12. Line 192: I. monospinus (neg = 11, pos = 6); I. asanumai (neg = 4, pos=5); I. nipponensis (pos = 4); I. persulcatus (neg = 4); and H. japonica (pos = 2).

- I would suggest to remove all results from the other tick species, since it does not fit the objective of the study: ‘This study describes the population genetic structure and gene flow of I. ovatus and H. flava and their Rickettsia infection rates along an altitudinal gradient’.

Response: Thank you for your suggestions. The authors agreed to focus on I.ovatus and H. flava data only, please see the revised L191-192:

The total number of positive (pos) and negative (neg) ticks for Rickettsia infections from the ticks collected in 2018 were: I. ovatus (neg = 22, pos = 17) and H. flava (neg = 10, pos = 13) .

We removed the information about the other tick species, please see L195-196:

In this study, the total number of samples from the previously published data [6;11] and the adult ticks whose cox1 was successfully sequenced per species were: I. ovatus (346) and H. flava (243). 

Table 1 was also revised and only the information about the I. ovatus and H.flava were retained. 

Also in L214-215 We detected SFGR in 78 (12.44%) out of 627 ixodid ticks, with the highest detected in I. ovatus (46/346; 13.29%) and in H. flava (15/243; 6.17%). 

We also removed in the discussion L343-363 In this study, only a few (n < 20) I. monospinus, I. asanumai, I. nipponensis and H. japonica were collected; however, a high Rickettsia infection rate (31.58%–100%) was found despite these limited numbers. In previous studies, high Rickettsia infection rates despite a few individuals were also observed in I. monospinus in several prefectures in Japan [71-73]. The extent of tick habitat distribution may vary between species, which may have influenced the widely different Rickettsia infection rates observed among the species in this study. The high Rickettsia infection rate in these ticks is probably due to effective transovarial transmission [74]. Furthermore, Rickettsial endosymbionts are inclined to have a high infection rate in some tick species populations because they contribute to the tick microbiome which might imply nutritional symbiosis in the tick species with high Rickettsia infection despite the low number of samples tested [75-77]. Additionally, the few collected individuals of these species may be habitat specialists that thrive in a narrow habitat range in contrast to I. ovatus and H. flava, which may be habitat generalists capable of thriving in a wide habitat range [78]. The narrow habitat range of habitat specialist ticks might have enabled increased interactions between infected and uninfected Rickettsia ticks within the small spatial scale compared to the habitat generalist ticks, thus causing their high Rickettsia infection rates. This conclusion should be considered with caution, since the limited number of individual ticks collected across the Niigata Prefecture may have caused a bias in the estimated Rickettsia infection rates determined in this study. We suggest conducting future research on Rickettsia infection rates in a wider sampling range with an increased number of I. monospinus, I. asanumai, I. nipponensis, and H. japonica individuals to further increase our understanding of the association between Rickettsiae and their tick vectors. 

13. PLOS authors have the option to publish the peer review history of their article (what does this mean?). If published, this will include your full peer review and any attached files.

Do you want your identity to be public for this peer review? For information about this choice, including consent withdrawal, please see our Privacy Policy.

No 

Response: The authors understand and respect the decision of Reviewer 2, to not disclose his/ her identity for peer review history once the manuscript is published.

14. From the attached file: Line 56: Ixodes ovatus (complete scientific name at beginning of the sentence). 

Response: The authors acknowledge this comment, and we revised this accordingly, please see L55-56

Ixodes ovatus populations had greater genetic divergence possibly…..

15. Line 84: Haemaphysalis flava to H. flava

Response: This was changed accordingly, please see L83: Japan in the following tick species: Dermacentor taiwanensis, H. flava, Haemaphysalis hystricis, …...

References: 

DIVA-GIS. n.d. Free Spatial Data. Accessed 2021. https://www.diva-gis.org/Data.

Farr, T.G., P.A. Rosen, E. Caro, R. Crippen, R. Duren, S. Hensley, M. Kobrick, et al. 2007. The shuttle radar topography mission: Reviews of Geophysics, v. 45, no. 2, RG2004. 

Google. 2021. Google Earth Engine Code Editor. Accessed 1 6, 2022. https://earthengine.google.com/platform/.

Gorelick, Noel Hancher, Matt Dixon, Mike Ilyushchenko, Simon Thau, David Moore, and Rebecca. 2017. "Google Earth Engine: Planetary-scale geospatial analysis for everyone." Remote Sensing of Environment (Elsevier) 202: 18-27. doi:10.1016/j.rse.2017.06.031.

Hijmans, Robert J., Luigi Guarino, and Prem Mathur. 2012. DIVA-GIS Manual Version 7.5. California.

---

## [Decision Letter · Decision Letter 1]

16 Oct 2023

PONE-D-23-12494R1Rickettsia infection rate along an altitudinal gradient as influenced by population genetic structure of Ixodes ovatus and Haemaphysalis flava ticksPLOS ONE

Dear Dr. Watanabe,

Thank you for submitting your manuscript to PLOS ONE. After careful consideration, we feel that it has merit but does not fully meet PLOS ONE’s publication criteria as it currently stands. Therefore, we invite you to submit a revised version of the manuscript that addresses the points raised during the review process.

We look forward to receiving your revised manuscript.

Kind regards,

Maria Stefania Latrofa

Academic Editor

PLOS ONE

Journal Requirements:

Reviewers' comments:

Reviewer's Responses to Questions

**Comments to the Author**

1. If the authors have adequately addressed your comments raised in a previous round of review and you feel that this manuscript is now acceptable for publication, you may indicate that here to bypass the “Comments to the Author” section, enter your conflict of interest statement in the “Confidential to Editor” section, and submit your "Accept" recommendation.

Reviewer #2: All comments have been addressed

Reviewer #3: (No Response)

2. Is the manuscript technically sound, and do the data support the conclusions?

Reviewer #2: Yes

Reviewer #3: Partly

3. Has the statistical analysis been performed appropriately and rigorously? 

Reviewer #2: Yes

Reviewer #3: Yes

4. Have the authors made all data underlying the findings in their manuscript fully available?

Reviewer #2: No

Reviewer #3: No

5. Is the manuscript presented in an intelligible fashion and written in standard English?

Reviewer #2: Yes

Reviewer #3: Yes

6. Review Comments to the Author

Reviewer #2: The manuscript was changed according to my suggestions. Although using already published data, all references were properly added, and authors reanalyzed and showed new results. The manuscript has a good flow of ideas, with all results presented in details. Also, ecological studies, although many times “suggestive”, are important for epidemiological information.

I would recommend another correction of the manuscript once the sequences are available at GenBank, so we can confirm and reanalyze the data. Only a few suggestions regarding the text.

Line: 114 ‘Genomic DNA was extracted using Isogenome DNA extraction kits (Nippon Gene Co.Ltd. Tokyo, Japan) following the manufacturer's recommended protocol.’

- Authors said that each tick was individually identified morphologically. Ticks were also extracted individually or in pools? Add please.

Table 1: ‘** not 195 enough samples for analysis’

- There is no ** on the table, only * and ***. Maybe it is a typing error?

Table 2: (indicated by ab). Please correct the ), it is not overwritten.

References

The references need standardization. Each reference is shown in a different format. Please check the author guidelines for proper standardization. Please be careful in the next submission, all references must be according to guidelines.

Data availability

Before the next ‘round’ of corrections, it would be necessary to control accession numbers. In the section “Material and Methods, Published data”, please add the accession numbers of the used sequences. They are already (I hope) available at GenBank, since it was already published. Also, in the section “Unpublished data”, add all accession numbers.

Reviewer #3: Overall, the authors have been receptive to the feedback and have made several changes in the manuscript to address the previous reviewers' concerns. However, the manuscript still echoes significantly with your previously published popgen study. The data and findings still appear to be an extension rather than presenting novel insights specific to Rickettsia. Below are specific suggestions for each section:

The title

"as influenced by population genetic structure" suggests a strong causal relationship between the genetic structure of the ticks and the Rickettsia infection rate. Also, there may be more than just the influence of the genetic structure on the infection rate (e.g., environmental factors, tick behavior, etc.). A suggestion: “Genetic Structure and Rickettsia Infection Rates in Ixodes ovatus and Haemaphysalis flava Ticks Across Different Altitudes"

Abstract

L25-26: Change to “The population genetic structure was analyzed utilizing the mitochondrial…”

L30-31: Change to “A significant difference was observed in Rickettsia infection rates and mean altitude per group between the two cluster groups and the three genetic groups identified within I. ovatus”

L33-34: I suggest a more cautious tone when making such conclusions eg. “Our results suggest a potential correlation between the low gene flow in I. ovatus populations and the spatially heterogeneous Rickettsia infection rates observed along the altitudinal gradient”

Introduction

L43-44: This sentence seems isolated, elaborate on why the size of ticks influences their dispersal and how host movement plays a role e.g. “Their dispersal is linked to the mobility of their hosts, relying on them to disperse into new landscapes and potentially introduce pathogens”

L83-44: This statement is overgeneralizing by saying “ticks” and “rickettsia”, try to be more specific.

L86-87: Add to this sentence the reasoning why this relationship is important to strengthen the introduction of your objectives. Can be reworded eg. “In this study, we elucidate the relationship between Rickettsia infection rates and population genetic structure along an altitudinal gradient…to improve public health etc.”

Methodology

L121: A table with PCR primers for rickettsia (like in Arai 2021), the target size and references and annealing temps used could be useful. Also try to reference the primary primer source for each rickettsial gene rather than secondary source (authors previously published work)

L175-178: It might be useful to provide more details about why these specific tests were chosen and if the data meet the assumptions of these tests (e.g. normality, equal variances etc.).

L130-135: The primer pair sequences for forward and reverse primers seems to be identical, please double check this info. Although you referenced your previous study for the details, include a reference for primer sequences as you’ve stated them with none.

L167-169: It’s not clear if a model testing tool was used to select the model used for the trees, if a model-testing tool was used but not mentioned, add a few sentences detailing this step as it will help improve stats robustness.

L170-177: After explaining the method, immediately tie it back to the implications it has for understanding Rickettsia infection rates or distribution.

Discussion

L256-257: use terms like “associated with” instead of “can cause” to ensure that the language reflects the type of relationship (causal or correlational) indicated by your data

Line 262-263: same as above point

L281-287: This is a strong point but could be strengthened. Discuss how your findings specifically align with adaptive evolutionary theory. Consider discussing any alternative explanations for the observed patterns and why local adaptation might be the most plausible explanation.

L288-290: Consider removing the emphasis of the previous study's results and instead highlight how these new samples provide additional insight or a different perspective on Rickettsia infection rates.

L303- 306: Consider also discussing how this limitation might specifically impact your findings

L310-327: Are there challenges/considerations that might need to be addressed to apply these findings in a real-world context?

GenBank Accession numbers: You mentioned uploading the new Cox sequences, what about the Rickettsia sequences as these are the focal point of the study (maybe I missed them while checking?) and are required for validation of your findings

Figures 1-4 : they are blurry and lack clarity, which might hinder the understanding of the data presented. guidelines by Plos one "Ensure that your images have a resolution of at least 300 pixels per inch (ppi) and appear sharp, not pixelated.

Be careful not to inadvertently reduce the resolution when creating a file in graphics editing software "

I hope the comments and suggestions are helpful with your paper.

Warm regards and happy revising!

7. PLOS authors have the option to publish the peer review history of their article (what does this mean?). If published, this will include your full peer review and any attached files.

Reviewer #2: No

Reviewer #3: No

---

## [Author Response · Author response to Decision Letter 1]

27 Dec 2023

2023 December 27

Dr. Maria Stefania Latrofa

Academic Editor

PLOS ONE

Dear Dr. Latrofa,

We are pleased to submit the revised version of our manuscript entitled "Genetic structure and Rickettsia infection rates in Ixodes ovatus and Haemaphysalis flava ticks across different altitudes". This resubmission incorporates all the modifications based on the valuable comments and suggestions provided by both the editors and reviewers.

Throughout the revision process, we carefully addressed each comment and suggestion received, ensuring a thorough response to improve the manuscript. We have made the necessary revisions in line with the editor's and reviewers' recommendations, and these changes have been indicated within the manuscript with their corresponding line numbers.

The authors agreed that this manuscript has not been published elsewhere nor is it under consideration by another journal. All authors have carefully reviewed and approved the revised content, aligning with PLOS ONE's submission policies. Additionally, we confirm that there are no conflicts of interest to disclose among the authors.

We revised the manuscript in adherence to PLOS ONE's style requirements and file naming. The authors are grateful to the editors and reviewers for their constructive comments and valuable suggestions, which have significantly improved the overall quality of the manuscript.

We hope that these revisions have improved the manuscript, making it suitable for publication in PLOS ONE journal. We look forward to your kind consideration and hope for a positive outcome.

Sincerely Yours, 

Dr. Kozo Watanabe, Professor

Center for Marine Environmental Studies (CMES) 

Ehime University, Bunkyo-cho 3, Matsuyama, 790-8577, Japan 

Email: watanabe.kozo.mj@ehime-u.ac.jp

Reviewer 2’s comments: Changes are highlighted in yellow in the revised manuscript with the track changes file

1. If the authors have adequately addressed your comments raised in a previous round of review and you feel that this manuscript is now acceptable for publication, you may indicate that here to bypass the “Comments to the Author” section, enter your conflict of interest statement in the “Confidential to Editor” section, and submit your "Accept" recommendation.

All comments have been addressed

Response: The authors are thankful for the reviewer’s comments and suggestions. 

2. Is the manuscript technically sound, and do the data support the conclusions?

Yes

Response: We are thankful for this comment. 

3. Has the statistical analysis been performed appropriately and rigorously?

Yes

Response: We appreciate your response. 

4. Have the authors made all data underlying the findings in their manuscript fully available?

No

Response: 

Please see L103-104 Sequences used for analysis are available in the GenBank database under the accession numbers MW063669 to MW064124 and MW065821 to MW066347.

For the unpublished data and additional samples, please see L113-114 The sequences are available in the GenBank database under the accession numbers OR975837 to OR975875 and OR975876 to OR975898.

5. Is the manuscript presented in an intelligible fashion and written in standard English?

Yes

Response: Thank you for the feedback. 

6. The manuscript was changed according to my suggestions. Although using already published data, all references were properly added, and authors reanalyzed and showed new results. The manuscript has a good flow of ideas, with all results presented in details. Also, ecological studies, although many times “suggestive”, are important for epidemiological information.

I would recommend another correction of the manuscript once the sequences are available at GenBank, so we can confirm and reanalyze the data. Only a few suggestions regarding the text.

Response: The authors appreciate all the comments and suggestions and we carefully addressed all of these in this revision. All the sequences have been deposited in Genbank as previously mentioned in the previous comment #4. 

7. Line: 122-123 ‘Genomic DNA was extracted using Isogenome DNA extraction kits (Nippon Gene Co.Ltd. Tokyo, Japan) following the manufacturer's recommended protocol.’

Authors said that each tick was individually identified morphologically. Ticks were also extracted individually or in pools? Add please.

Response: We revised it accordingly please see L122-123: Genomic DNA was extracted from individual ticks using Isogenome DNA extraction kits (Nippon Gene Co. Ltd. Tokyo, Japan) following the manufacturer's recommended protocol.

8. Table 1: ‘** not 195 enough samples for analysis’

- There is no ** on the table, only * and ***. Maybe it is a typing error?

Response: The authors removed L203: ** not enough samples for analysis. Instead, we retained L204: **tick species identification is based on molecular identification using the cox1 marker and BLAST results

9. Table 2: (indicated by ab). Please correct the ), it is not overwritten.

Response: The authors removed the ), please see L258: cluster dendrogram groups 1 and 2 indicated by ab

10. References

The references need standardization. Each reference is shown in a different format. Please check the author's guidelines for proper standardization. Please be careful in the next submission, all references must be according to guidelines.

Response: Each reference was checked and formatted into PLOS one reference style, Vancouver. We have updated the list of references and its corresponding line in the manuscript. 

11. Data availability

Before the next ‘round’ of corrections, it would be necessary to control accession numbers. In the section “Material and Methods, Published data”, please add the accession numbers of the used sequences. They are already (I hope) available at GenBank, since it was already published. Also, in the section “Unpublished data”, add all accession numbers.

Response:

Please see L103-104 Sequences used for analysis are available in the GenBank database under the accession numbers MW063669 to MW064124 and MW065821 to MW066347.

For the unpublished data and additional samples, please see L113-114 The sequences are available in the GenBank database under the accession numbers OR975837 to OR975875 and OR975876 to OR975898.

 

Reviewer 3’s comments: Changes are highlighted in green in the revised manuscript with the track changes file.

1. If the authors have adequately addressed your comments raised in a previous round of review and you feel that this manuscript is now acceptable for publication, you may indicate that here to bypass the “Comments to the Author” section, enter your conflict of interest statement in the “Confidential to Editor” section, and submit your "Accept" recommendation.

 (No Response)

Response: The authors respect the reviewer’s opinion. 

2. Is the manuscript technically sound, and do the data support the conclusions?

Partly

Response: The authors are thankful for the comments and suggestions of the Reviewer. 

3. Has the statistical analysis been performed appropriately and rigorously?

Yes 

Response: We appreciate your response 

4. Have the authors made all data underlying the findings in their manuscript fully available?

No

Response: We have addressed this comment as previously mentioned in Reviewer 2, no. 4. Please see L103-104 Sequences used for analysis are available in the GenBank database under the accession numbers MW063669 to MW064124 and MW065821 to MW066347.

For the unpublished data and additional samples, please see L113-114 The sequences are available in the GenBank database under the accession numbers OR975837 to OR975875 and OR975876 to OR975898.

5. Is the manuscript presented in an intelligible fashion and written in standard English?

Yes

Response: The manuscript has been checked thoroughly for any English grammatical errors by an English proofreading company for scientific manuscripts. 

6. Overall, the authors have been receptive to the feedback and have made several changes in the manuscript to address the previous reviewers' concerns. However, the manuscript still echoes significantly with your previously published popgen study. The data and findings still appear to be an extension rather than presenting novel insights specific to Rickettsia. Below are specific suggestions for each section:

Response: Thank you the authors have carefully addressed each of the comments and suggestions of the reviewers and editors. 

7. The title

"as influenced by population genetic structure" suggests a strong causal relationship between the genetic structure of the ticks and the Rickettsia infection rate. Also, there may be more than just the influence of the genetic structure on the infection rate (e.g., environmental factors, tick behavior, etc.). A suggestion: “Genetic Structure and Rickettsia Infection Rates in Ixodes ovatus and Haemaphysalis flava Ticks Across Different Altitudes"

Response: We revised the title as suggested, please see L1-2 Full Title: Genetic structure and Rickettsia infection rates in Ixodes ovatus and Haemaphysalis flava ticks across different altitudes

Abstract

8. L25-26: Change to “The population genetic structure was analyzed utilizing the mitochondrial…”

Response: Please see L26-27 “The population genetic structure was analyzed utilizing the mitochondrial cytochrome oxidase 1 (cox1) marker.”

9. L30-31: Change to “A significant difference was observed in Rickettsia infection rates and mean altitude per group between the two cluster groups and the three genetic groups identified within I. ovatus”

Response: The authors agreed to this suggestion, please see L31-33: A significant difference was observed in Rickettsia infection rates and mean altitude per group between the two cluster groups and the three genetic groups identified within I. ovatus.

10. L33-34: I suggest a more cautious tone when making such conclusions eg. “Our results suggest a potential correlation between the low gene flow in I. ovatus populations and the spatially heterogeneous Rickettsia infection rates observed along the altitudinal gradient”

Response: We revised it accordingly, please see L34-36 “Our results suggest a potential correlation between the low gene flow in I. ovatus populations and the spatially heterogeneous Rickettsia infection rates observed along the altitudinal gradient.”

Introduction

11. L43-44: This sentence seems isolated, elaborate on why the size of ticks influences their dispersal and how host movement plays a role e.g. “Their dispersal is linked to the mobility of their hosts, relying on them to disperse into new landscapes and potentially introduce pathogens”

Response: Kindly see L45-47: Their dispersal is linked to the mobility of their hosts, relying on them to disperse into new landscapes and potentially introduce pathogens [7-8].

12. L83-44: This statement is overgeneralizing by saying “ticks” and “rickettsia”, try to be more specific.

Response: We appreciate this comment thus we revised the sentence into, please see L86-89: “To our knowledge, no previous studies have considered the influence of environmental factors on the spatial distribution of Spotted fever group Rickettsia infection rates along an altitudinal gradient in local Ixodid tick populations such as Ixodes ovatus and Haemaphysalis flava as influenced by the tick population’s genetic structure.”

13. L86-87: Add to this sentence the reasoning why this relationship is important to strengthen the introduction of your objectives. Can be reworded eg. “In this study, we elucidate the relationship between Rickettsia infection rates and population genetic structure along an altitudinal gradient…to improve public health etc.”

Response: Please see L90-92: “In this study, we elucidate the relationship between Rickettsia infection rates as influenced by population genetic structure along an altitudinal gradient to improve public health understanding of the distribution of ticks and tick-borne diseases.

Methodology

14. L121: A table with PCR primers for rickettsia (like in Arai 2021), the target size and references and annealing temps used could be useful. Also try to reference the primary primer source for each rickettsial gene rather than secondary source (authors previously published work)

Response: The authors have agreed to this suggested thus we included a supplementary table to show the PCR primers, target size, annealing temperature and its corresponding references in Supplementary Table 2 (S2 Table). 

Please see the revised L131: … and and outer membrane protein B gene (rOmpB) as described and analyzed in [6,37-41] (S2 Table) and L667-668: S2 Table. Summary of PCR primers used in the detection of Spotted fever group Rickettsia 

15. L175-178: It might be useful to provide more details about why these specific tests were chosen and if the data meet the assumptions of these tests (e.g. normality, equal variances etc.).

Response: Kindly see L185-190: To determine whether there was a significant difference in the Rickettsia infection rate between haplotype groups for I. ovatus and H. flava, we performed a z-score test at p < 0.05. The z-score test was chosen because of the large sample size and because the population variance was known. To determine whether there were differences in the mean altitude between the haplotype groups, we used the Welch t-test at p < 0.05. Welch t-test was used when the means of the two populations were normally distributed and had equal variances.

16. L130-135: The primer pair sequences for forward and reverse primers seems to be identical, please double check this info. Although you referenced your previous study for the details, include a reference for primer sequences as you’ve stated them with none.

Response: Thank you for this comment, we revised it carefully please check L139-140: LCO-1490 (5′- GGTCAACAAATCATAAAGATATTGG-3') and HCO1–2198 (5′– AAACTTCAGGGTGACCAAAAAATCA- 3) for phylogenetic analysis and tick species identification [42].

17. L167-169: It’s not clear if a model testing tool was used to select the model used for the trees, if a model-testing tool was used but not mentioned, add a few sentences detailing this step as it will help improve stats robustness

Response: Kindly see L174-176: Briefly, we constructed a Bayesian phylogenetic tree of cox1 haplotypes for I. ovatus and H. flava, respectively, using Markov chain Monte Carlo (MCMC) approach implemented in the BEAST version 1.10.14 [43].

18. L170-177: After explaining the method, immediately tie it back to the implications it has for understanding Rickettsia infection rates or distribution.

Response: We have included this in the manuscript, please see L172-175: We constructed a haplotype network analysis using PopART program version 1.7 (http://popart.otago.ac.nz/index.shtml) on cox1 I. ovatus and H. flava sequences to assess the haplotype relationships and the distribution of Rickettsia infected ticks using the median-joining network algorithm [47]. 

Also L185-187: To determine whether there was a significant difference in the Rickettsia infection rate between haplotype groups for I. ovatus and H. flava, we performed a z-score test at p < 0.05.

Discussion

19. L256-257: use terms like “associated with” instead of “can cause” to ensure that the language reflects the type of relationship (causal or correlational) indicated by your data

Response: We agreed to these suggestions, thus we revised L268-271: Our findings support our hypothesis that a genetically structured tick population, such as I. ovatus is associated with the Rickettsia infection rate to be spatially heterogenous due to limited gene flow along an altitudinal gradient.

20. Line 262-263: same as above point

Response: Kindly see L272-274: Despite the addition of new samples of I. ovatus and H. flava, we found a similar pattern of population genetic structure from the previous study of [11] thus supporting the robustness of their population genetic structure results. 

21. L281-287: This is a strong point but could be strengthened. Discuss how your findings specifically align with adaptive evolutionary theory. Consider discussing any alternative explanations for the observed patterns and why local adaptation might be the most plausible explanation.

Response: Please see L293-301: “The different Rickettsia infection rates and altitudinal ranges between the I. ovatus phylogenetic groups may be caused by diverse factors such as host availability and distribution, other environmental factors such as climate and vegetation, and anthropogenic factors such as urbanization. However, the adaptive evolutionary theory, which states that organisms adjust to new or severe changes in their environment to become better suited to their habitat [59-60] best explains our results. Based on the relationship between the I. ovatus phylogenetic groups and their mean attitudes, I. ovatus might be undergoing local adaptation along the altitudinal gradient due to the higher genetic differentiation between populations as supported by the significant global FST (0.4154) found in I. ovatus.”

22. L288-290: Consider removing the emphasis of the previous study's results and instead highlight how these new samples provide additional insight or a different perspective on Rickettsia infection rates.

Response: The authors agreed into this comment, thus please see L301-305: Based on isolation by environment (IBE), genetic differentiation will increase with increased environmental differences independent of geographic distances [33-34;61]. Thus in our study, I.ovatus collected across an altitudinal gradient have shown genetic differences and different Rickettsia infection rates. 

23. L303- 306: Consider also discussing how this limitation might specifically impact your findings

Response: We have revised these sentences and incorporated your suggestions, please see L319-324: One of the limitations of this study is the use of one mitochondrial gene cox1 which limited us to compare our results with other target genes to highly support our findings. If markers with high mutation rates or many markers were used, it might have been possible to look at even finer population genetic structure and see differences in infection rates among the subdivided populations. Despite this, we were able to determine the relationship between the tick population genetic structure and Rickettsia infection rates as influenced by the altitudinal gradient. The mitochondrial cox1 gene has been widely used for population genetic analysis of many tick species and was proven to be informative in determining the relationship from the subfamily to the population levels [87-91].

24. L310-327: Are there challenges/considerations that might need to be addressed to apply these findings in a real-world context?

Response: The authors appreciate these suggestions thus we revised it accordingly, please see L328-339: In future studies, we suggest including additional mitochondrial genes and or nuclear genes.

Since ticks are blood-sucking ectoparasites, they directly influence their mammalian hosts and the pathogens they transmit [76-78]. The interaction between the vector (tick), host, and pathogen (Rickettsia) is essential in understanding and predicting the risk and transmission of tick-borne diseases [79]. Understanding the genetic structure of ticks can serve as an alternative indicator to infer the potential spread of its pathogen [80]. Our study found relationships between (1) the population genetic structure of ticks and the corresponding Rickettsia infection rates, (2) altitude and the population genetic structure of ticks, and (3) altitude and Rickettsia infection rates. Though our results can provide useful information about the tick distribution and possible potential spread of pathogens, some factors should also be considered to apply our results in the field setting such as ticks can have different mammalian hosts during different life stages that have varying hosts mobility and other environmental factors such as temperature, humidity etc.. can also be a factor. 

25. GenBank Accession numbers: You mentioned uploading the new Cox sequences, what about the Rickettsia sequences as these are the focal point of the study (maybe I missed them while checking?) and are required for validation of your findings

Response: We have addressed this comment as previously mentioned in Reviewer 2, no. 4. Please see L103-104 Sequences used for analysis are available in the GenBank database under the accession numbers MW063669 to MW064124 and MW065821 to MW066347.

For the unpublished data and additional samples, please see L113-114 The sequences are available in the GenBank database under the accession numbers OR975837 to OR975875 and OR975876 to OR975898.

26. Figures 1-4 : they are blurry and lack clarity, which might hinder the understanding of the data presented. guidelines by Plos one "Ensure that your images have a resolution of at least 300 pixels per inch (ppi) and appear sharp, not pixelated.

Be careful not to inadvertently reduce the resolution when creating a file in graphics editing software "

Response: The authors have edited the figures to obtain a clearer and higher resolution, thank you for these suggestion. 

27. I hope the comments and suggestions are helpful with your paper.

Warm regards and happy revising!

Response: The authors are thankful for all the comments and suggestions of the reviewers and editor which greatly improved our manuscript for publication. 

28. PLOS authors have the option to publish the peer review history of their article (what does this mean?). If published, this will include your full peer review and any attached files.Do you want your identity to be public for this peer review? For information about this choice, including consent withdrawal, please see our Privacy Policy.

No

Response: The authors respect the decision of the Reviewer to be anonymous.

---

## [Editor Report · Decision Letter 2]

30 Jan 2024

Genetic structure and Rickettsia infection rates in Ixodes ovatus and Haemaphysalis flava ticks across different altitudes

PONE-D-23-12494R2

Dear Dr. Watanabe,

We’re pleased to inform you that your manuscript has been judged scientifically suitable for publication and will be formally accepted for publication once it meets all outstanding technical requirements.

Kind regards,

Maria Stefania Latrofa

Academic Editor

PLOS ONE

---

## [Editor Report · Acceptance letter]

3 Mar 2024

PONE-D-23-12494R2 

PLOS ONE

Dear Dr. Watanabe, 

I'm pleased to inform you that your manuscript has been deemed suitable for publication in PLOS ONE. Congratulations! Your manuscript is now being handed over to our production team.

Kind regards, 

on behalf of

Dr. Maria Stefania Latrofa 

Academic Editor

PLOS ONE